# Non-selective distribution of infectious disease prevention may outperform risk-based targeting

Benjamin Steinegger[1], Iacopo Iacopini [2,3], Andreia Sofia Teixeira [4,5], Alberto Bracci [6], Pau Casanova-Ferrer [7,8], Alberto Antonioni[7] & Eugenio Valdano [9✉]

Epidemic control often requires optimal distribution of available vaccines and prophylactic tools, to protect from infection those susceptible. Well-established theory recommends prioritizing those at the highest risk of exposure. But the risk is hard to estimate, especially for diseases involving stigma and marginalization. We address this conundrum by proving that one should target those at high risk only if the infection-averting efficacy of prevention is above a critical value, which we derive analytically. We apply this to the distribution of pre-exposure prophylaxis (PrEP) of the Human Immunodeficiency Virus (HIV) among men-having-sex-with-men (MSM), a population particularly vulnerable to HIV. PrEP is effective in averting infections, but its global scale-up has been slow, showing the need to revisit distribution strategies, currently risk-based. Using data from MSM communities in 58 countries, we find that non-selective PrEP distribution often outperforms risk-based, showing that a logistically simpler strategy is also more effective. Our theory may help design more feasible and successful prevention.

[1] Departament d'Enginyeria Informàtica i Matemàtiques, Universitat Rovira i Virgili, Tarragona, Spain. [2] Department of Network and Data Science, Central European University, Vienna, Austria. [3] Aix Marseille Univ, Université de Toulon, CNRS, CPT, Marseille, France. [4] LASIGE, Departamento de Informática, Faculdade de Ciências, Universidade de Lisboa, Lisboa, Portugal. [5] INESC-ID, Lisboa, Portugal. [6] Department of Mathematics, City, University of London, London, UK. [7] Grupo Interdisciplinar de Sistemas Complejos (GISC), Department of Mathematics, Carlos III University of Madrid, Leganés, Spain. [8] Department of Systems Biology, Centro Nacional de Biotecnología, CNB-CSIC, Madrid, Spain. [9] Sorbonne Université, INSERM, Institut Pierre Louis d'Epidémiologie et de Santé Publique, Paris, France. ✉email: eugenio.valdano@inserm.fr

Pre-exposure prophylaxis (PrEP) of the Human Immuno-deficiency Virus (HIV) is the use of antiretroviral medications to prevent HIV acquisition, by uninfected individuals. More than ten years have passed since the first evidence that PrEP could protect people from HIV[1], and PrEP is now a component of the HIV prevention cascade[2]. Its uptake, however, has been restricted to a few countries[3], and is currently inadequate in the context of the global effort toward eliminating the HIV/AIDS epidemic[2,3]. A challenge to PrEP scale-up is its distribution, which requires identifying potential candidates, supplying the medication, and providing the necessary follow-up to ensure consistent use. Most guidelines[4] and cost-effectiveness studies[5–8] recommend offering PrEP to those at high risk of acquiring HIV[9,10]. Risk, however, is difficult to measure, and particularly so among those who would benefit the most from PrEP, as it is the case of men-having-sex-with-men (MSM)[4,7,11]: Stigma and punitive laws often marginalize communities and make them hard-to-reach[12]. Proposed metrics for estimating individual risk often exhibit poor accuracy[4,11], or maybe operationally too challenging[13], when faced with real-world complexity[14]. Profiling risk may also reinforce stigma[7].

Risk-based distribution strategies apply to diseases other than HIV, and types of prevention other than chemoprophylaxis. It is the case with many vaccines[15]: early vaccination of healthcare workers against COVID-19 is the latest notable example. One underpinning of risk-based distribution is that high infection risk comes, at least partially, from having many contacts with other individuals through which the infection may be acquired. Once infected, however, having many contacts means having a high probability of further spreading the pathogen, i.e., of causing superspreading events. Thus, targeting those at high risk means preventing superspreading events, preventing a large number of infections, and lowering incidence in the population. In the formalism of complex networks, whereby nodes are individuals and links are contacts along which the pathogens can spread[16], this means prioritizing the highest-degree nodes (hubs)[17,18]. Many extensions to this theory have appeared[19–24], but the main tenet has remained the same: you should protect those who can cause superspreading events, if infected. Prevention strategies that target individuals with specific contact patterns, however, require detailed information on the underlying network structure, which is hard to get[25,26], and thus not part of routine surveillance, as the case of PrEP among MSM shows[8]. As a result, these strategies may perform well in models, but are hard to translate into public health guidelines.

Our study helps to bypass this limitation, by proving that targeting those at high risk of causing superspreading events may not be the best-performing strategy in all settings: Simpler strategies, which are easier to implement, maybe more effective. We do that by studying the role of the individual-level efficacy of prevention: Efficacy measures how well prophylaxis, or vaccination, protects the recipient from infection[27]. 100% efficacy means that those who use prevention cannot be infected; below this value, efficacy is the probability that prevention averts a transmission event that would otherwise occur. We demonstrate that efficacy determines which distribution strategy works best in reducing community-level disease circulation, and that targeting those at highest risk is optimal only if efficacy is above a threshold, which we derive analytically. We also find that PrEP efficacy is below this threshold in many MSM communities in the world. In these communities, non-selective PrEP distribution likely outperforms targeted distribution, showing that the logistically simplest distribution strategy is also the most effective.

## Results

We start from the observation that, if efficacy is below 100%, higher risk of exposure to the pathogen entails a higher chance that prevention fails, leading to a breakthrough infection[28–30]. This specifically concerns hubs, as the number of contacts determines—at least partially—the risk of exposure. We thus posit the existence of a trade-off. On the one hand, standard theory tells us that protecting those with many contacts brings down population-level transmission, given that they can cause superspreading events, when infected. On the other hand, their chance of experiencing breakthrough infections may be high.

We quantitatively investigate the existence, and phenomenology, of this trade-off, using the heterogeneous mean-field formalism[16] on an annealed network with degree distribution $p(k)$. Each node in the network has degree $k$ sampled from $p(k)$, establishing $k$ contacts (links) with other nodes. Heavy-tailed degree distributions are typically used to model heterogeneity in the number of contacts[31]. We assume here that node degrees along links are not correlated. Real contact networks may however exhibit assortative behavior[31]: high-degree nodes tend to be in contact with high-degree nodes. In Supplementary Note 1, we cover the case of assortative networks. Annealed networks are particularly suitable when the timescale of pathogen spread is much larger than the timescale at which contacts change[16], as is the case of HIV epidemics in MSM communities (Supplementary Note 2). Also, annealed networks can be parametrized from existing surveys[5,13,32], unlike more complex network models, which would require high-resolution contact data. To describe disease spread, we use the Susceptible-Infectious-Susceptible compartmental model[16], by which a susceptible individual becomes infected at a rate $\lambda$, when in contact with an infectious individual. Also, those infected spontaneously transition to the susceptible state at rate $\mu$. This last process may model recovery, or population turnover, as in the case of HIV infection[5] (see also Supplementary Note 3). We also assume leaky[27] prevention, with efficacy $\epsilon$ in decreasing the instantaneous transmission rate: $\epsilon = 1$ corresponds to maximally effective prevention.

The heterogeneous mean-field formalism is a customary approach to write the equations describing the evolution in time of the spread of the disease in terms of the probability, by degree class, that a node is infected[16]. It can deal with arbitrary degree distributions, while factoring out all dynamical correlations in the status of connected nodes, which would render the theory intractable. In our case, these equations are

$$\begin{cases} \dot{x}_k = -\mu x_k + \frac{\lambda}{\langle k \rangle} k(1 - x_k)\xi \\ \dot{y}_k = -\mu y_k + \frac{\lambda}{\langle k \rangle}(1 - \epsilon)k(1 - y_k)\xi \\ \xi = \sum_k k p_k \big[(1 - g_k)x_k + g_k y_k\big]. \end{cases} \quad (1)$$

Here, $x_k$ is the probability that an individual in degree class $k$ who does not receive prevention is infected, $y_k$ is the probability that an individual in degree class $k$ who receives prevention is infected, $\lambda$ is the transmission rate, $\mu$ is the recovery rate, $g_k$ is the probability that an individual in degree class $k$ receives prevention. $\xi$ is an auxiliary variable that encodes the probability of establishing a contact with an infected individual. It is the extension, in the case of a partially immunized population, of the customary coupling term of the heterogeneous mean-field equations[16]. The form of $\xi$ given in Eq. (1) implies no degree-degree correlations: see Supplementary Note 1 for nonzero assortativity. For convenience, we also define the reduced transmission rate as $\hat{\lambda} = \lambda/(\langle k \rangle \mu)$, where $\langle k \rangle$ is the average degree.

**Optimal distribution of prevention.** Community-level prevalence can be written as a function of the quantities in Eq. (1): $I[g, x, y] = \sum_k p_k\big[(1 - g_k)x_k + g_k y_k\big]$. We now wish to derive the impact that increasing prevention among a specific degree class

has on decreasing community-level prevalence, for different values of efficacy $\epsilon$. Optimizing the distribution strategy is relevant when large-scale distribution and adoption is not possible. We thus start from the configuration of no prevention ($g = 0$), and study the impact of providing prevention to a small number of individuals, in degree class $k$, by means of the following linear response function: $f(k) = -(1/p_k)\mathrm{d}I/\mathrm{d}g_k\big|_{g=0}$ (see Methods for a detailed explanation). If protecting hubs is the best-performing strategy, then $f$ will be a monotonously increasing function of $k$. The existence of the trade-off will instead be marked by the existence of maximum of $f$, at finite $k$. At the endemic equilibrium ($\dot{x}_k = \dot{y}_k = 0$), we derive the expression of $f$ from Eq. (1) (see Methods):

$$f(k) = \overbrace{(x_k - y_k)\big|_{g=0}}^{F_{dir}(k)} + \overbrace{\frac{1}{p_k}\sum_m p_m \frac{dx_m}{dg_k}\bigg|_{g=0}}^{F_{indir}(k)}. \tag{2}$$

$f$ has two terms. $F_{dir}(k)$ quantifies the direct reduction in risk of infection among those receiving prevention. $F_{indir}(k)$ quantifies the indirect effect of the prevention campaign: the reduction in risk of infection among those who did not receive prevention, due to the presence of those who did. We derive the expression of both terms:

$$F_{dir}(k) = \frac{\epsilon \hat{\lambda} z k}{\left[1 + \hat{\lambda} z k\right]\left[1 + (1-\epsilon)\hat{\lambda} z k\right]}, \tag{3}$$

$$F_{indir}(k) = \frac{\psi \hat{\lambda}}{1 - \phi} k F_{dir}(k), \tag{4}$$

where $\hat{\lambda}, \psi, \phi$ depend on the degree distribution and the epidemic parameters, but do not depend on $\epsilon, k$. Their detailed expressions are provided in the Methods, along with the details of the calculation (see also Supplementary Note 4). Supplementary Note 5 shows the agreement of the analytical derivation with the numerical counterpart. We first examine the direct effect. $F_{dir}$ has a maximum when $k = k^*_{dir} \sim 1/\sqrt{1-\epsilon}$. If protection is perfect ($\epsilon = 1$), then $F_{dir}$ is monotonously increasing, meaning that highly connected individuals should always be prioritized, as the current theory mandates. If $\epsilon < 1$, $k^*_{dir}$ is finite (Fig. 1a), and above it, the high chance of breakthrough infection among high-risk individuals offsets the direct gain in protecting those who are most likely to get infected. We now turn to the indirect effect. $F_{indir}$ increases monotonously for any value of efficacy $\epsilon$ (Fig. 1a). This means that providing prevention to highly connected individuals always induces the largest indirect benefit on those who are not using prevention. The combination of $F_{dir}$ and $F_{indir}$ gives the following optimal degree for degree-prioritized prevention strategies:

$$k^* = \frac{1 + \sqrt{\left(\frac{z(1-\phi)}{\psi} - 1\right)\left(\frac{z(1-\phi)}{\psi}(1-\epsilon) - 1\right)}}{\hat{\lambda} z \left[\frac{z(1-\phi)}{\psi}(1-\epsilon) - (2-\epsilon)\right]}. \tag{5}$$

$k^*$ is always greater than $k^*_{dir}$, as it is the result of the effect of direct protection, which is optimal at $k^*_{dir}$, and indirect protection, which increases with $k$ (Fig. 1b, c). Furthermore, while $k^*_{dir}$ is finite whenever protection is non perfect ($\epsilon < 1$), $k^*$ is finite if $\epsilon < \epsilon_c \le 1$, with

$$\epsilon_c = \frac{(1-\phi)z - 2\psi}{(1-\phi)z - \psi}. \tag{6}$$

Equation (6) is our main theoretical result: There exists a value of critical efficacy, which is analytically computable, and which discriminates between the two following parameter regions. In the *high-efficacy region* ($\epsilon \ge \epsilon_c$) you should prioritize those at risk of causing superspreading events. In the *low-efficacy region* ($\epsilon < \epsilon_c$), instead, targeting individuals in degree class $k = k^* < \infty$ has the strongest impact on community prevalence.

Notably, the location and shape of the two parameter regions depend on baseline endemic prevalence (i.e., the prevalence in the absence of prevention). Highly prevalent diseases have higher $\epsilon_c$, and, in the low-efficacy region, lower $k^*$ (see Methods for the proof). This means that the same prevention tool (fixed $\epsilon$) may warrant different distribution strategies in different settings (Fig. 2a). In particular, individuals with many contacts should be targeted in low-prevalence communities ($\epsilon_c \le \epsilon$). Contrarily, where prevalence is high ($\epsilon_c > \epsilon$), they should not.

This also implies that the invasion phase of an epidemic is always in the high-efficacy region: if the aim of the prevention campaign is to minimize the likelihood of an outbreak of a disease which is not yet circulating, rather than eliminating an endemic disease, then targeting those at risk of causing superspreading events will always be optimal. This can be seen as the zero-prevalence limit of the above derivation, or more rigorously by computing the epidemic threshold[16], as we do in the Methods.

The value of the critical efficacy $\epsilon_c$ depends on network topology, too. Specifically, more heterogeneous contact networks have higher $\epsilon_c$. This is shown in Fig. 2b where, in the case of negative binomial degree distribution, $\epsilon_c$ increases as overdispersion increases. Supplementary Note 4 reports the same

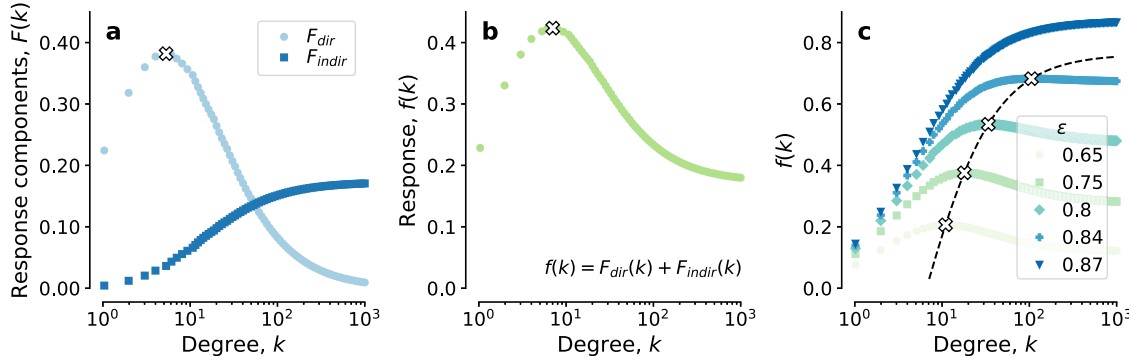

**Fig. 1 The response function $f$ and its components. a** Terms $F_{dir}(k)$ [Eq. (3)] and $F_{indir}(k)$ [Eq. (4)] of the response function. The cross indicates $k^*_{dir}$, maximum of $F_{dir}$. Reduced transmissibility is $\hat{\lambda} = 0.25$; degree distribution is a negative binomial with mean 2.0, coefficient of variation (standard deviation over mean) 4.7; efficacy is $\epsilon = 0.5$. **b** Response function $f(k)$, which is the sum of the terms in **a**. The cross indicates $k^*$, maximum of $f$. **c** Response function $f(k)$ for different values of $\epsilon$. Dashed line and crosses indicate $k^*$. Reduced transmissibility is $\hat{\lambda} = 2$; degree distribution is a negative binomial with mean 2.0, coefficient of variation 4.7.

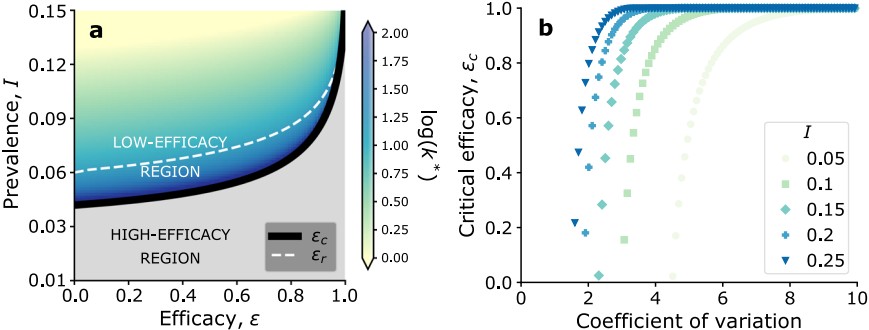

**Fig. 2 Critical efficacy, and the low-efficacy region. a** $k^*$ as a function of efficacy $\epsilon$, and baseline disease prevalence (prevalence without prevention). Finite values of $k^*$ mark the low-efficacy region. The gray area indicates $k^* \to \infty$: the high-efficacy region. The solid black lines shows critical efficacy $\epsilon_c$; the dashed black line shows $\epsilon_r$. The degree distribution is a negative binomial with mean 2.0, coefficient of variation 9.4. The reduced transmissibility $\hat{\lambda}$ is numerically set to match the corresponding baseline prevalence. **b** Critical efficacy $\epsilon_c$ as a function of the heterogeneity of the contact network, measured as the coefficient of variation of a negative binomial distribution with mean 2.0. Different curves mark different values of baseline prevalence.

result for a power-law degree distribution. Degree-degree correlations also increase the critical efficacy $\epsilon_c$, and, in the low-efficacy region, decrease $k^*$ (see Supplementary Note 1). Intuitively, this happens because assortativity increases risk of exposure among those already at high risk, exacerbating the likelihood of breakthrough infections.

We remark that efficacy ($\epsilon$) measures the level of leakage of prevention, i.e., how well it brings down the chance of transmission upon contact[27]. It should not be confused with the all-or-none mode of action[27], by which some instruments of prevention may completely fail to protect some individuals. This latter mechanism does not change the relative effectiveness of different distribution strategies, and is therefore not a factor in our study.

The low-efficacy region features, by definition, a class of individuals with finite degree $k^*$, that should be prioritized. This is conceptually consequential, but it may have a limited operational impact. We thus investigate whether, when $\epsilon < \epsilon_c$, offering prevention non selectively (random targeting) still outperforms targeting those at risk of causing superspreading events. A new critical value – $\epsilon_r$ – emerges, and splits the low-efficacy region in two. When efficacy is lower than $\epsilon_c$, but higher than $\epsilon_r$, targeting individuals with degree $k^*$ is still the best-performing strategy, but targeting those at risk of causing superspreading events outperforms non-selective targeting. The opposite is true when, instead, efficacy is lower than $\epsilon_r$. We call transition zone the part of the low-efficacy region where $\epsilon > \epsilon_r$: there, the choice of the distribution strategy is strongly determined by the practical constraints on being able to identify individuals at given levels of risk (see Fig. 2a). Degree-degree correlations in the contact network have the effect of lowering $\epsilon_r$ (see Supplementary Note 1).

**Pre-exposure prophylaxis of HIV.** We now apply this theory to PrEP in MSM communities, which is a prime candidate for exhibiting the emergence of a low-efficacy region, for several reasons. First, it is generally recommended in high-prevalence settings, and MSM has 25 times greater risk of acquiring HIV than heterosexual men[2]. Second, the efficacy of PrEP varies widely. It depends on the regimen (daily[1] v on-demand[33]), and on the level of adherence to the regimen: generally, efficacy falls in the range 40–90%[34]. Plus, the protection PrEP provides is leaky[34], as resistance to tenofovir/emtricitabine—the most common oral PrEP formulation—is rare[35]. Finally, the distribution of the number of sexual interactions an individual has (degree distribution) is heterogeneous[32,36].

We used estimates of HIV prevalence among MSM, coverage of antiretroviral treatment, prevalence of viral suppression, to compute the effective prevalence, i.e., the fraction of individuals at risk of transmitting HIV. We did it for 58 countries, and 24 cities. We used a negative binomial degree distribution with empirically-informed parameters. Data sources and details on the numerical estimations are available in Supplementary Note 6.

We set PrEP efficacy to 60%, and found that 34 out of 78 communities are in the high-efficacy region, 44 in the low-efficacy region. Among the latter, 4 are in the transition zone. Europe is in the high-efficacy region, in accordance with previous studies recommending risk-based distribution[5,37]. Notably, many communities in areas of active PrEP roll-out[38] are in the low-efficacy region: it is the case of Brazil and of those in southern Africa (excluding Botswana). These results have four implications. First, different communities may warrant different PrEP distribution strategies, as they find themselves in different parameter regions. This provides corroborating evidence to the current international commitment to eliminating the HIV/AIDS epidemic through geographically tailored responses and interventions[39]. Second, effective interventions require epidemiological and behavioral data at high accuracy and resolution, whose collection is also at the center of international efforts, at least programmatically[39]. High accuracy ensures that the region (low-efficacy vs high-efficacy) is correctly estimated, high resolution responds to the fact that spatially contiguous communities may be epidemiologically different: Fig. 3b shows several examples of the latter phenomenon. One is Cameroon, which national estimates put in the high-efficacy region, but epidemiological data from two of its cities, Douala and Yaoundé, point to the low-efficacy region. Namibia and Botswana are another example: they are neighboring countries, which share generalized HIV epidemics, but lie in different parameter regions. Third, parameter region assignment is weakly sensitive to PrEP efficacy. We chose the mid-range value of 60%. Lowering it to the value of the IprEx trial[1] (44%) would cause only 2 out of 76 communities to change parameter region (Fig. 3c). Increasing it to that of the IPERGAY study[33] (86%) would cause 5 out of 76 countries to change region (Fig. 3d). This ensures that our assignment is robust across PrEP efficacy estimates.

We also tested the impact of assortative mixing, which is reported in many MSM communities. Namely, location-based partner selection[40], and homophily[8], may cause those at high risk to mix preferably with other high-risk individuals. Assortativity had little effect on the efficacy estimates of Fig. 3b: Specifically, with the assortativity estimated in Ref.[41], only 4 out of 34 communities moved from the high-efficacy region to the

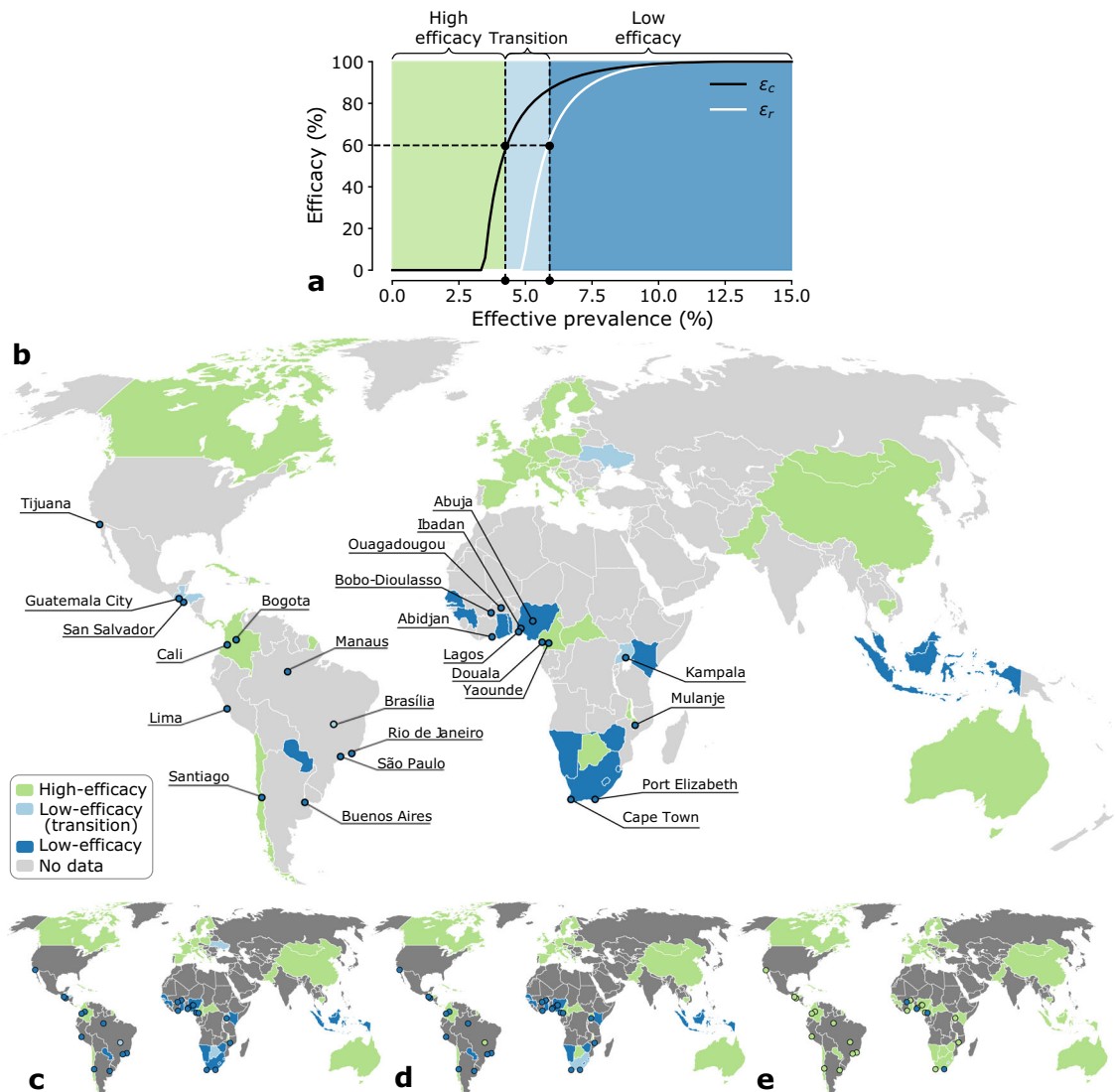

**Fig. 3 PrEP distribution in communities of men-having-sex-with-men (MSM) in 58 countries. a** Phase diagram of PrEP in MSM communities. The x-axis shows the effective prevalence, i.e., the fraction of individuals who are living with HIV and can potentially transmit it. It is estimated from data on HIV prevalence, treatment coverage, viral suppression rate (see Supplementary Note 6 and Supplementary Note 7). The y-axis shows efficacy of PrEP. The horizontal dashed line is 60% efficacy. The solid black curve is critical efficacy $\epsilon_c$, the solid white curve is $\epsilon_r$. The range of effective prevalence in the high-efficacy region at 60% efficacy is colored in green. The range of effective prevalence in the low-efficacy region at 60% efficacy is colored in dark blue, and light blue (transition zone). **b** Map showing parameter region estimates in 58 countries, 24 cities (see Supplementary Note 6 for data sources and estimates of effective prevalence). Communities in the high-efficacy region are green, communities in the low-efficacy region are in dark blue, light blue (transition zone). **c** The same as **b**, assuming efficacy at 44% (IprEx trial). **d** The same as **b**, assuming efficacy at 86% (IPERGAY study). **e** The same as **b**, in the scenario that each community reaches UNAIDS's 95–95–95 targets for testing, treatment, and viral suppression.

transition zone, with risk-based distribution still outperforming non-selective distribution (see Supplementary Note 1). We also checked a value of assortativity twice as much as that of Ref. [41] (see Supplementary Note 1): in that case, 7 of 34 communities moved from the high-efficacy region to the transition zone, and 2 out of 4 moved from the transition zone to the low-efficacy region. This shows that assortativity may change recommendations for PrEP distribution only at extremely high values (i.e., those at high risk strongly favoring mixing with others at high risk), and for only 2 out of the 76 communities investigated here.

Generally, populations at low coverage tend to be in the low-efficacy region (see Supplementary Note 7). In these communities, our results corroborate the calls to shift the focus away from risk[7]. As treatment coverage expands, communities may then transition to the high-efficacy region, showing that the scale-

up of treatment and of prevention should be in sync: As treatment expands, prevention should adapt. To exemplify this, we investigated what would happen if UNAIDS's 95-95-95 targets for testing, treatment, and viral suppression were reached[39]. Figure 3 shows that 38 out of 44 communities that are now in the low-efficacy region would transition to the high-efficacy region. Notably, however, many communities in Africa would remain in the low-efficacy parameter region even at that extremely high treatment coverage.

Stigma and criminalization of same-sex acts are other factors possibly associated with the low-efficacy region: They are obstacles to PrEP use, as they make it harder to supply the medication, and to provide consistent support and follow-up. This decreases adherence, which in turns decreases efficacy. Decriminalization and societal changes leading to lower stigma

may thus signal a transition from the low-efficacy to the high-efficacy region.

Finally, the availability of new PrEP formulations may affect the conditions and timing of the transition to the high-efficacy region. Notably, long-acting injectable cabotegravir (CAB-LA) was recently shown to have higher efficacy than oral PrEP[42]. This means that communities that are now in the low-efficacy region for oral PrEP, maybe in the high-efficacy region for CAB-LA.

## Discussion

We set up a theoretical formalism to identify the best strategy for population-level distribution of primary prevention. We found that the infection-preventing efficacy of prevention, disease prevalence, and the underlying contact structure determine under which conditions nonselective distribution of prevention outperforms risk-based distribution.

We then applied it to pre-exposure prophylaxis of HIV among men-having-sex-with-men. Non-selective PrEP distribution is effective when HIV prevalence is high and/or treatment coverage is low. Then, as prevalence goes down and treatment increases, focusing on protecting individuals at the highest risk will likely become the best-performing strategy. At the same time, more consistent use of oral PrEP, or new long-acting PrEP formulations may speed up the progression to the high-efficacy region. When this happens, it is possible that many communities will find themselves in the transition zone, at least temporarily. There, risk-based distribution should already be favored over non-selective distribution, as in the high-efficacy region.

Our work has limitations. We focused on optimizing the reduction of community-level disease burden, and did not consider other aspects of primary prevention, such as providing equitable access to prevention, or improving the quality of life of marginalized individuals. Our study did not include factors which can influence risk of acquisition: in the case of HIV, we did not explicitly account for the effect of primary prevention other than PrEP[2]. Specifically, whereas our framework does account for an arbitrary overall rate of condom use by means of the transmissibility parameter $\lambda$, it does not include possible changes in condom use among those on PrEP, due to possible behavioral adaptation[43]. The compartmental model we used is a coarse-grained representation of the progression of HIV infection, and its transmission. In particular, it does not account for the different transmission probability of receptive and insertive anal sex. This, however, would potentially bias our findings only if PrEP use were consistently correlated with type of act (insertive vs receptive). Summing up, more detailed HIV models, and community-specific estimates of partner selection patterns, could provide better numerical estimates of critical efficacy, and thus be useful in applied studies focusing on specific communities. We also remark that our study applies to the infection-preventing effect of medications. Some of them also reduce morbidity and mortality among the infected, as it is the case of vaccines against COVID-19. As such, the main criterion for their distribution has been the risk of developing severe disease, which is beyond the scope of our study. Also, we show in Supplementary Note 8 that the low-efficacy parameter region of COVID-19 vaccination requires very low vaccine efficacy, or unrealistically high incidence, finding no evidence against risk-based distribution. Finally, our model does not consider the fact that targeting high-risk individuals may be inevitable, if the side effects of the medication or vaccine outweigh its benefits only when the probability to be exposed to the pathogen is high.

HIV prevention is but one public health challenge in which the low-efficacy region may be present: Whenever vaccination campaigns aim at reducing incidence in high-prevalence settings, estimating the value of critical efficacy could help optimize vaccine distribution. It might be the case, for instance, of Plasmodium falciparum malaria, as a new vaccine formulation may soon become available[44]. When this happens, adapting our theory to malaria – both in terms of vector-borne transmission and mixing network[45] – will help inform roll-out, especially where parasite prevalence is high.

## Methods

### The linear response function $f$

*Definition.* The goal of $f(k)$ is to measure the impact that providing prevention to a few individuals in degree class $k$ has on community-level baseline prevalence. We assume a population of $N$ individuals, and define $f$ as the change in the number of infected individuals in the population ($NI$), due to a small change in the amount of prevention provided in degree class $k$ ($Np_k g_k$):

$$f(k) \sim -\frac{\mathrm{d}(NI)}{\mathrm{d}(Np_k g_k)}\bigg|_{g=0}, \tag{7}$$

where the minus sign is due to the fact that prevention will bring prevalence down. Here $I$ is community-level prevalence as defined previously. Then, the population size $N$ correctly cancels out (the final result does not depend on population size), and we get to the final definition of the response function:

$$f(k) = -\frac{1}{p_k}\frac{\mathrm{d}I}{\mathrm{d}g_k}\bigg|_{g=0}, \tag{8}$$

*Derivation of* $F_{dir}$. At the endemic equilibrium ($\dot{x}_k = \dot{y}_k = 0$), one can use Eq. (1) to write $y_k$ as a function of $x_k$:

$$y_k = \frac{1-\epsilon}{1-\epsilon x_k}x_k. \tag{9}$$

Also, given that $x_k$ has to be evaluated at $g = 0$, we can use its recursive form, which comes from setting $g = 0$ in the first line of Eq. (1):

$$x_k = \frac{z\hat{\lambda}k}{1+z\hat{\lambda}k}, \tag{10}$$

where $z = \langle kx \rangle$. Angle brackets denote expectation values on the degree distribution, so in this case this would mean

$$z = \langle kx \rangle = \sum_k p_k k x_k. \tag{11}$$

$z$ has a clear epidemiological interpretation, as it measures the expected number of at-risk contacts that an individual makes. Specifically, $z = \langle k \rangle l$, where $l$ is the probability that a given contact is with an infected individual. This measure is sensitive to the amount of heterogeneity in the network. Indeed, if the network had a homogeneous degree distribution (i.e., all individuals had degree close to $\langle k \rangle$), then $z \approx \langle k \rangle I$ ($I$ is the prevalence as usual). Broad degree distributions give instead $z > \langle k \rangle I$, meaning that the probability of establishing a contact with an infected individual is higher than the probability of finding an infected individual at random in the population.

Plugging Eqs. (9)–(10) into Eq. (2), one gets Eq. (3).

*Derivation of* $F_{indir}$. In the following, we implicitly assume that all should be evaluated at $g = 0$. At equilibrium ($\dot{x} = 0$), we perform the derivative $\frac{d}{dg_m}$ on both sides of the first line in Eq. (1):

$$-\frac{dx_k}{dg_m} + \hat{\lambda}k\left[-\frac{dx_k}{dg_m}z + (1-x_k)\frac{d\xi}{dg_m}\right] = 0. \tag{12}$$

We compute the derivative of $\xi$ from its definition in Eq. (1):

$$\frac{d\xi}{dg_m} = mp_m(y_m - x_m) + \sum_{k'} k' p_{k'}\frac{dx_{k'}}{dg_m}, \tag{13}$$

and insert it into Eq. (12):

$$\sum_{k'}\left[\hat{\lambda}k(1-x_k)k'p_{k'} - \delta_{kk'}(1+z\hat{\lambda}k)\right]\frac{dx_{k'}}{dg_m} = -\hat{\lambda}k(1-x_k)mp_m(y_m-x_m). \tag{14}$$

This equation constitutes a linear system for the matrix $J_{km} = dx_k/dg_m$. Defining the auxiliary variables $u_k = \hat{\lambda}k(1-x_k)$, $v_k = kp_k$, $w_k = kp_k(y_k - x_k)$ and $D_{kk'} = (1+z\hat{\lambda}k)\delta_{kk'}$, we can rewrite Eq. (14) as

$$(\mathbf{uv}^T - \mathbf{D})\mathbf{J} = -\mathbf{uw}^T. \tag{15}$$

To get $\mathbf{J}$, we note that the matrix $\mathbf{uv}^T - \mathbf{D}$ is a rank-1 perturbation of a diagonal matrix, and invert it by means of Ref. [46] (Sherman–Morrison formula):

$$\mathbf{J} = \frac{1}{1 - \mathbf{v}^T\mathbf{D}^{-1}\mathbf{u}}\mathbf{D}^{-1}\mathbf{uw}^T. \tag{16}$$

By inserting the definitions of $\mathbf{u}, \mathbf{v}, \mathbf{w}$ and $\mathbf{D}$ into Eq. (16), and after some algebra, we get an explicit expression of $\mathbf{J}$, and thus the derivative $dx_k/dg_m$.

Now, with $dx_k/dg_m$, $y_k$, and $x_k$ at hand, and again after some algebra, we get to the final form of $F_{indir}$ [Eq. (4)], and thus $f(k)$:

$$f(k) = \frac{\epsilon \hat{\lambda} z k}{(1 + \hat{\lambda} z k)\left[1 + (1-\epsilon)\hat{\lambda} z k\right]}\left(1 + \frac{\hat{\lambda}\psi}{1-\phi}k\right), \quad (17)$$

where we defined

$$\phi = \hat{\lambda}\left\langle\left(\frac{k}{1+z\hat{\lambda}k}\right)^2\right\rangle \text{ and } \psi = \left\langle\frac{k}{(1+z\hat{\lambda}k)^2}\right\rangle. \quad (18)$$

Expectation values are computed similarly to Eq. (11) (see also Supplementary Note 9).

*Critical point of f.* The derivative of $f(k)$ in Eq. (17) is proportional to the following:

$$f'(k) \sim k^2\hat{\lambda}^2 z\left[\psi(1-\phi)(2-\epsilon) + z(1-\epsilon)\right] - 2\hat{\lambda}\psi(1-\phi)k + 1. \quad (19)$$

In the above expression, we dropped a strictly positive term that multiplies the rhs. Evaluating the derivative at $k = 0$, we immediately see that $f'(0) > 0$. Accordingly, a sufficient condition for $f(k)$ to have a maximum in $\mathbb{R}_+$ is $\lim_{k\to\infty} f'(k) < 0$. For large $k$, the leading term is the quadratic one in Eq. (19). Therefore, the condition $\lim_{k\to\infty} f'(k) < 0$ requires the quadratic term to be positive, i.e.

$$z(1-\phi) - 2\psi > \epsilon\left[z(1-\phi) - \psi\right]. \quad (20)$$

From their definitions, we know that $z, \phi, \psi > 0$. Let us now assume that the term on the right hand side (RHS) is negative. In this case, the above condition would read

$$\frac{z(1-\phi) - 2\psi}{z(1-\phi) - \psi} < \epsilon. \quad (21)$$

The variable $\epsilon$ is bounded between [0, 1]. Therefore, the condition in Eq. (21) can only be fulfilled if the left hand side is smaller than one. However, this is impossible since it would require $2\psi < \psi$. Thus, for $k^*$ to exist, the RHS in Eq. (20) must be positive, which necessarily requires $\phi < 1$. Accordingly, Eq. (20) can be written as

$$\epsilon < \epsilon_c = \frac{z(1-\phi) - 2\psi}{z(1-\phi) - \psi}. \quad (22)$$

It is straightforward to show that $\epsilon_c < 1$. Further, $\epsilon_c$ is positive if $z(1-\phi) > 2\psi$. Eventually, solving for $f'(k) = 0$ then gives $k^*$ as in Eq. (5).

**Effect of prevalence and network heterogeneity on $k^\star, \epsilon_c$.** $z$ is sensitive both to prevalence, and to network heterogeneity. In particular, if prevalence increases, then $z$ increases (given that $z \geq \langle k\rangle I$). Also, $z$ increases if network heterogeneity increases, too. This happens because, in more heterogeneous networks, higher-degree nodes will more likely be infected than low-degree ones. We can prove this rigorously in the case of power-law-distributed degrees. From Eqs. (10)-(11), the following equation for $z$ follows:

$$\hat{\lambda}\sum_k p_k \frac{k^2}{1 + z\hat{\lambda}k} = 1. \quad (23)$$

Assuming $p_k = (\gamma - 1)k^{-\gamma}$, and approximating sums on $k$ with integrals, this equation becomes

$${}_2F_1\left(1, \gamma - 2, \gamma - 1, -\frac{1}{z\hat{\lambda}}\right) = z\frac{\gamma - 2}{\gamma - 1}, \quad (24)$$

where ${}_2F_1$ is the ordinary hypergeometric function. This has a simple pole at $\gamma = 2$, and there, ${}_2F_1(1, \gamma - 2, \gamma - 1, -\frac{1}{z\hat{\lambda}}) \approx \frac{1}{z(\gamma - 2)}$. In the vicinity of $\gamma = 2$, Eq. (24) thus becomes

$$\frac{1}{\left[z(\gamma - 2)\right]^2} \approx \frac{1}{\gamma - 1}. \quad (25)$$

The rhs is finite in $\gamma = 2$. This implies $z \sim 1/(\gamma - 2)$ to kill the divergence in the lhs. Hence, $z$ becomes larger as the network becomes more heterogeneous (i.e., $\gamma$ decreases towards $\gamma = 2$).

When instead the network becomes more homogeneous (i.e., $\gamma$ becomes larger and larger), Eq. (24) tends to

$$z \approx 1 - \frac{1}{\hat{\lambda}}; \quad (26)$$

which is exactly its lower bound ($z = \langle k\rangle I$, as previously discussed). This completes the proof that $z$ increases when either prevalence increases, or the network becomes more heterogeneous.

Now, when $z$ is large, the following approximate relations hold:

$$\phi \approx \frac{1}{z^2\hat{\lambda}}; \quad (27)$$

$$\psi \approx \frac{\langle k^{-1}\rangle}{z^2\hat{\lambda}^2}. \quad (28)$$

This implies that Eqs. (5)–(6), in the $z \to \infty$ limit, become

$$k^* \approx \frac{1}{\hat{\lambda}z}\frac{1 + \sqrt{1-\epsilon}}{1-\epsilon} \to 0; . \quad (29)$$

$$\epsilon_c \approx 1 - \frac{\langle k^{-1}\rangle}{z^3\hat{\lambda}^2} \to 1; \quad (30)$$

proving that higher prevalence, and higher network heterogeneity, cause $\epsilon_c$ to increase, and, in the low-efficacy region, cause $k^*$ to decrease.

**Invasion stage and epidemic threshold.** To calculate the epidemic threshold of the system, we study the stability of the disease-free equilibrium ($x_k = y_k = 0$), as customary[16].

First, we linearize Eq. (1) around $x_k = y_k = 0$:

$$\begin{cases} \dot{x}_k = -\mu x_k + \frac{\lambda}{\langle k\rangle}k\xi \\ \dot{y}_k = -\mu y_k + \frac{\lambda}{\langle k\rangle}(1-\epsilon)k\xi. \end{cases} \quad (31)$$

From these, we derive an equation for $\xi$, by multiplying the first line by $kp_k(1 - g_k)$, the second line by $kp_k g_k$, and then sum them together, and sum over $k$. This gives

$$\dot{\xi} = \left[-\mu + \frac{\lambda}{\langle k\rangle}\left(\langle k^2\rangle - \langle gk^2\rangle\right)\right]\xi. \quad (32)$$

The disease-free equilibrium is no longer stable for transmissibility values giving $-\mu + \frac{\lambda}{\langle k\rangle}\left(\langle k^2\rangle - \langle gk^2\rangle\right) > 0$. This gives the following epidemic threshold:

$$\lambda_c = \lambda_0\left(1 - \frac{\epsilon\langle gk^2\rangle}{\langle k^2\rangle}\right)^{-1}. \quad (33)$$

Here, $\lambda_0$ is the well-known value of the epidemic threshold in the absence of prevention ($g = 0$): $\lambda_0 = \mu\langle k\rangle/\langle k^2\rangle$[47]. We now define the response function for the epidemic threshold:

$$f_\lambda(k) = \frac{1}{p_k}\frac{d\lambda_c}{dg_k}\bigg|_{g=0}, \quad (34)$$

Unlike Eq. (8), here there is no minus sign because prevention increases the epidemic threshold. With some algebra, one gets:

$$f_\lambda(k) = \frac{\epsilon\lambda_0}{\langle k\rangle}k^2, \quad (35)$$

which is monotonously increasing in $k$, proving that the invasion stage of the epidemic is always in the high-efficacy region.

**Transition zone and $\epsilon_r$.** We measure the impact of non-selective distribution (random targeting) as the expected value of $f(k)$ over the degree distribution: $\langle f\rangle$. We compare it with the impact of targeting those at risk of causing superspreading events, as $f(\infty) = \lim_{k\to\infty} f(k)$. The former – $\langle f\rangle$ – must be evaluated numerically. The latter is easy to derive from Eqs. (3)-(4):

$$f(\infty) = \frac{\psi}{z(1-\phi)}\frac{\epsilon}{1-\epsilon}. \quad (36)$$

The critical value $\epsilon_r$ is the efficacy value for which $\langle f\rangle = f(\infty)$.

**Reporting summary.** Further information on research design is available in the Nature Research Reporting Summary linked to this article.

## Data availability
Estimates of HIV prevalence and treatment coverage as discussed in Supplementary Note 6 and Supplementary Note 7 are available from the cited references in the Supplementary Information, and from UNAIDS at https://aidsinfo.unaids.org/ (accessed February 2022).

## Code availability
The code used in this study is available here: https://github.com/steinegg/non_selective_distribution_prophylaxis[48].

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

## Acknowledgements

We acknowledge the Complexity72h workshop, held at IMT School in Lucca, Italy, 17–21 June 2019, where this study was conceived. B.S. acknowledges financial support from the European Unions Horizon 2020 research and innovation program under the Marie Skłodowska-Curie Grant Agreement No. 713679 and from the Universitat Rovira i Virgili (URV). I.I. acknowledges support from the James S. McDonnell Foundation 21st Century Science Initiative Understanding Dynamic and Multi-scale Systems - Post-doctoral Fellowship Award and from the Agence Nationale de la Recherche (ANR) project DATAREDUX (ANR-19-CE46-0008). A.S.T acknowledges support from FCT and the LASIGE and INESC-ID Research Units, refs: UIDB/00408/2020, UIDP/00408/2020, PTDC/EEI-SII/1937/2014, and UIDB/50021/2020. A.A. gratefully acknowledges the financial support of the Spanish Ministry of Science and Innovation under grant n. IJC2019-040967-I. P.C.F acknowledges financial support from the by the Spanish Ministerio de Ciencia, Innovación y Universidades (MICINN) under Projects FIS2016-78313-P and PID2019-109320GB-100/AEI/10.13039/501100011033. The funders had no role in study design, data collection and analysis, decision to publish, or preparation of the manuscript. The authors are grateful to Sally Blower and Vittoria Colizza for useful feedback.

## Author contributions

B.S. and E.V. conceived of, designed the study, and performed the theoretical calculations. B.S. set up and carried out the numerical calculations. B.S., I.I., A.S.T., A.B., P.C.F., A.A., E.V. discussed the results. I.I. designed the figures. E.V. wrote the manuscript. B.S. wrote the Supplementary Information. B.S., I.I., A.S.T., A.B., P.C.F., A.A., E.V. revised the manuscript.

## Competing interests

The authors declare that they have no competing interests.
