## [Peer Review File · Nature Communications]

Non-selective distribution of infectious disease prevention may outperform risk-based targetingREVIEWER COMMENTS

Reviewer #1 (Remarks to the Author):

In this paper, the authors use a mathematical model to show that targeting prevention to those with most contacts may not always be most effective in reducing prevalence of an infection in the population. This might have implications for the way PrEP is distributed in some countries with a high prevalence of HIV, because targeting is not always practically feasible and non-targeted distribution may be much easier to realize. This is an interesting question and would also intuitively make sense.

While I appreciate this novel approach to thinking about prevention, I have some major concerns about the model presented, namely the following:

1. If I understand the model correctly, it assumes random mixing by degree. This means that high risk individuals (or superspreaders) do not have an increased risk for having contact with other high risk individuals. This is a very strong assumption, which is not mentioned explicitly in the paper. In classical theory, from which the approach of targeting prevention to the high risk group (the so-called core group) comes, this is exactly the reason why targeting is successful. High risk individuals have an increased rate of contacting other high risk individuals, therefore forming a core group, in which continued transmission can take place. Targeting this group has a disproportionately large impact on transmission, because it reduces not only the individual risk of a susceptible of becoming infected, but also the risk of transmission to other high risk individuals. This core group effect is neglected in the model presented here.

For MSM populations, the core group effect is strong, because high risk individuals meet each other in specific locations, and not just randomly, partly also because there is stigmatization and meeting locations are therefore limited.

Assumptions about contact patterns in the population should be made more clear and discussed already when the model is introduced.

2. The authors say that using an SI framework for HIV simply means that the recovery rate can be interpreted as population turnover. However, there is a major difference between these processes. Recovery rates in the population depend on the prevalence, i.e. higher prevalence means also more recoveries per time unit. However, this should not be the case for population turnover, where more deaths from a disease would not automatically be replaced by births. The demographic process in the model is not really explained and only appears implicitly.

3. In the introduction the authors say that they are using a complex networks approach. However, in networks there would be dependency between the contacts of a node, i.e. in their infection status. This is not taken into account here.

Minor comments:

Lines 78-93: Please add some explanation of the function ξ

After equation (4): explain what z is. Is $\Phi < 1$? Are Φ and $\Psi > 0$?

Equation (6): How can ϵ_c be ≤ 1 ? From equation (6) I conclude $\epsilon_c \geq 1$.

I find the terminology "high efficacy phase" a bit confusing, because I with the word "phase" I connect something that changes over time. I suggest the term "high prevention efficacy".

Lines 116-118: this needs some more explanation. How does the underlying contact structure play a role here? Maybe it would be good to mention the contact structure already when the model is introduced, and say how it influences the prevalence.

Line 192: "these" instead of "this"

Line 197: Unclear how treatment is included in the model. Treatment would lower the infectiousness of those under treatment or would move them back to the susceptible state. This is now not included in the model.

Line 235: Here the number of individuals using prevention is mentioned, but up to there the model is formulated in terms of fractions, and also the prevalence I as defined in line 85 is a fraction. This seems inconsistent to me. Please explain exactly how I is defined as a function of numbers of people taking prevention.

Reviewer #2 (Remarks to the Author):

The manuscript deals with the interesting idea that protecting the individuals with the highest contact rates is not always the best strategy to reduce the prevalence of an endemic disease when prevention is not 100% effective (leaky prevention). The situation the authors have in mind is the use of pre-exposure prophylaxis (PrEP) in control HIV and they apply their results using HIV epidemiological data from MSM communities in 58 countries and 24 cities.

To prove this claim, the epidemic spread is modelled using a heterogeneous mean-field SIS model with two types of infected individuals: those who do not receive prevention and those who do. The goal is then to find the optimal distribution of PrEP among individuals with different degrees that maximizes the initial reduction of the prevalence when PrEP are distributed in a population where no preventive measures are adopted.

As far as I know, the manuscript offers a new approach to deal with endemic infectious diseases and presents new insights of significance to the field of epidemics. The paper is, in general, well written but some revision is needed.

First, the explanation of the model could be better if it were done with a general audience in mind, i.e., readers unfamiliar with network theory. In particular, an interpretation of $\langle k \rangle$ (or, better, $\langle k \rangle_c$) would help to better understand of the model because it is the key ingredient. Moreover, it is important to mention the assumptions of this formulation (for instance, it is assumed that nodal degrees are uncorrelated).

Second, the main theoretical result -Eq.(6)-, is given in the main text without the meaning/interpretation of any of the ingredients, even though at least one of them (z) is easily interpretable. From a modelling point of view, the meaning and/or properties of the averages $\langle \psi \rangle$ and $\langle \phi \rangle$, the other two ingredients, seem to be very important. For instance, if $\langle \phi \rangle \geq 1$, the optimal degree k^* given by Eq. (5) is negative. In such a case, what does $k^* < 0$ mean? Is this hypothesis on $\langle \phi \rangle$ feasible?

On the other hand, if $\langle \phi \rangle < 1$, then $k^* > 0$ if the denominator is positive but, in this case, the critical efficacy given by the right-hand side of Eq. (6) will be negative if $2\langle \psi \rangle > z(1-\langle \phi \rangle)$ (the positivity of the denominator of k^* already guarantees that $z(1-\langle \phi \rangle) > \langle \psi \rangle$). So, I think that a more detailed discussion about the conditions that guarantee $k^* > 0$ and critical efficacy < 1 would be very helpful.

Moreover, there are very important claims like "highly prevalent diseases have higher $\langle \epsilon_c \rangle$, and, in the low-efficacy phase, lower k^* " and "more heterogeneous contact networks have higher $\langle \epsilon_c \rangle$ " that are not justified. Are these statements obtained only from the numerical simulation of the model? Can they be proved from the definition of $\langle \epsilon_c \rangle$ and the properties of $\langle \psi \rangle$, $\langle \phi \rangle$, and z ? What is the relationship between these quantities and disease prevalence?

Finally, it would be interesting to confirm numerically that the prevalence in the new endemic equilibrium that emerges after the introduction of preventive measures actually follows the predictions based on the linear response function.

Other comments:

Why a negative binomial degree distribution is used as a main example? It is well known that distributions of contacts with high variance (negative exponential, truncated scale-free distribution) are more suitable to model contact patterns among individuals? Moreover, the example of COVID-19 vaccination considers such a degree distribution with a mean degree equal to 1, which seems to be quite unrealistic.

The definition of $F_{dir}(k)$ given in (2), namely, $(y_k - x_k)|_{g=0}$, leads to Eq. (3) multiplied by -1 when it is used in combination with Eqs. (9) and (10). I think its definition should be $(x_k - y_k)|_{g=0}$.

The writing of the section "Invasion stage and epidemic threshold" has to be carefully revised (see last minor comments below).

In summary, although the manuscript is interesting and offers novel insights, I cannot recommend its publication in its current form but I encourage the authors to improve the writing.

Minor comments:

- The use of the name "diffusion equations" to refer to the system of ODEs governing the dynamics of the disease is misleading and it is not the usual one.

- Eq.(10) directly follows from the equilibrium equation for (1) after replacing ξ by z since $\xi(g=0) = z$. I don't see why Ref.[42] is needed.

- Line 103: $F_{indir}(k)$ does not decrease but increases with k .

- Line 108: "... it is the result of..."

- Line 133: Is the sentence " ϵ_c decreases as overdispersion increases" correct? In Fig.2b, ϵ_c increases with the coefficient of variation.

- Line 249: I think the use of "by" in the expression "we derive the first line in Eq (1) by g_m " is not correct in English.

- Check the left-hand side of Eq.(14). The second term in parentheses must be $(1 + \hat{\lambda} z k)$

- Line 267: Give more detail in the sentence "The condition $\lim_{k \rightarrow \infty} f'(k) < 0$ leads to ϵ_c of Eq.(6)."

- The linearization about the disease-free state of the first equation in (1) is equal to the right-hand side of Eq.(20) multiplied by μ .

- Check the expression (21) of Θ , in particular, the factor $\langle k \rangle$ in the sum.

- Line 274: Specify the variable with respect to which the derivative is to be taken.

- The denominator in the fraction appearing in (23) has to be $\langle k^2 \rangle$ instead of $\langle k \rangle$.

Reviewer #3 (Remarks to the Author):

This manuscript explores the impact of current targeted PrEP recommendations, with a discussion of when non-selective PrEP distribution may be optimal. I have a few questions/recommendations for the authors:

- The authors discuss alternative dosing strategies and adherence, but now long-acting injectable PrEP is available in certain countries. How could this impact calculations and estimates?
- I wonder whether the authors have considered some important characteristics in HIV risk beyond PrEP adherence and number of sexual interactions. If not, these should be included in the Limitations section. Specifically, it would be useful to at least mention:
 - Differences in per-act risk (receptive vs. insertive anal sex)
 - Prevalence of condom usage (which could help counter poor adherence)
 - Impact of criminalization of same-sex behavior. For example, sexual negotiation for MSM in Nigeria is very different than in Canada.
 - Role of homophily in transmission.
- Stating that non-selective PrEP distribution is optimal is helpful, but I would like the authors to go a bit further with their recommendations. What does this look like? Several thresholds are mentioned for high vs. low efficacy - what is recommended for the transition from non-selective to targeted?

TITLE

Non-selective distribution of infectious disease prevention may outperform risk-based targeting

Point-by-point response to reviewers' comments

Reviewer #1

In this paper, the authors use a mathematical model to show that targeting prevention to those with most contacts may not always be most effective in reducing prevalence of an infection in the population. This might have implications for the way PrEP is distributed in some countries with a high prevalence of HIV, because targeting is not always practically feasible and non-targeted distribution may be much easier to realize. This is an interesting question and would also intuitively make sense.

While I appreciate this novel approach to thinking about prevention, I have some major concerns about the model presented, namely the following:

1. If I understand the model correctly, it assumes random mixing by degree. This means that high risk individuals (or superspreaders) do not have an increased risk for having contact with other high risk individuals. This is a very strong assumption, which is not mentioned explicitly in the paper. In classical theory, from which the approach of targeting prevention to the high risk group (the so-called core group) comes, this is exactly the reason why targeting is successful. High risk individuals have an increased rate of contacting other high risk individuals, therefore forming a core group, in which continued transmission can take place. Targeting this group has a disproportionately large impact on transmission, because it reduces not only the individual risk of a susceptible of becoming infected, but also the risk of transmission to other high risk individuals. This core group effect is neglected in the model presented here.

For MSM populations, the core group effect is strong, because high risk individuals meet each other in specific locations, and not just randomly, partly also because there is stigmatization and meeting locations are therefore limited.

This is a very good point, and we thank you for bringing this up. We have now extended our analysis to include degree-degree assortativity, whereby those with many contacts tend to be in contact with other individuals with many contacts. This translates into high-risk individuals being preferably in contact with other high-risk individuals. In the SI we now include a section (Supplementary Note 4) in which we derive the efficacy thresholds ϵ_c , ϵ_r , and the optimal degree k^* , for arbitrary assortativity. The approach mixes analytical derivation, and numerical evaluation of formulas which cannot be simplified analytically.

We find that assortativity increases the critical efficacy ϵ_c , increases ϵ_r , and, in the low-efficacy region, decreases the optimal degree k^* . This happens because assortativity increases risk of exposure among those already at high risk (many contacts). This in turn increases the probability of getting infected despite prevention.

We then apply these new findings to the case of PrEP, and re-assign the communities in Fig. 3 to the different efficacy regions, assuming a value of assortativity that was reported for Sweden (Fig. S4a,b). Assortativity causes no change in region assignment for most communities: only 4 go from the high-efficacy region (assuming no assortativity), to the transition zone (with assortativity). Plus, this entails no change in strategy, given that both the high-efficacy region and the transition zone risk-based distribution is recommended. We also tested a value of assortativity twice as high as that, to explore communities with mixing patterns which might be very different from the Swedish study, which can be hardly regarded as representative of different socioeconomic and epidemiological contexts. In this case, compared to the analysis in the main text (no assortativity), 7 communities moved from the high-efficacy region to the transition zone, and 2 communities moved from the transition zone to the low-efficacy zone. This shows that assortativity may change recommendations for PrEP distribution only at extremely high values (i.e., those at high risk strongly favoring mixing with others at high risk), and for few communities. Notwithstanding, further studies applied to specific settings would certainly benefit from community-specific estimates of partner selection patterns (and the resulting assortativity), in order to accurately estimate ϵ_c and ϵ_r , in case assortativity turns out to be very different from the values tested.

We have added the following paragraphs to the main text:

1.

We quantitatively investigate the existence, and phenomenology, of this trade-off, using the heterogeneous mean field formalism on an annealed network with degree distribution $p(k)$. Each node in the network has degree k sampled from $p(k)$, establishing k contacts (links) with other nodes. Heavy-tailed degree distributions are typically used to model heterogeneity in the number of contacts. We assume here that node degrees along links are not correlated. Real networks may however exhibit assortative behavior: high-degree nodes tend to be in contact with high-degree nodes. In Supplementary Note 4 we cover the case of assortative networks.

2.

Degree-degree correlations also increase the critical efficacy ϵ_c , and, in the low-efficacy region, decrease k^ (see Supplementary Note 4). Intuitively, this happens because assortativity increases risk of exposure among those already at high risk, exacerbating the likelihood of breakthrough infections.*

3.

We also tested the impact of assortative mixing, which is reported in many MSM communities. Namely, location-based partner selection, and homophily, may cause those at high risk mixing preferably with other high-risk individuals. Assortativity had little effect on the efficacy estimates of Fig. 3b: Specifically, with the assortativity estimated in Ref. 4, only 4 out of 34 communities moved from the high-efficacy region to the transition zone, with risk-based distribution still outperforming non-selective distribution (see Supplementary Note

4). We also checked a value of assortativity twice as much as that of Ref. 4 (see Supplementary Note 4): in that case, 7 of 34 communities moved from the high-efficacy region to the transition zone, and 2 out of 4 moved from the transition zone to the low-efficacy region. This shows that assortativity may change recommendations for PrEP distribution only at extremely high values (i.e., those at high risk strongly favoring mixing with others at high risk), and for only 2 out of the 76 communities investigated here.

The new section in the SI (Supplementary Note 4) describes the computation of $f(k)$, and illustrates the effect of assortativity through the following figures:

(a) Terms $F_{dir}(k)$ and $F_{indir}(k)$ for different values of assortativity, ω . (b) Response function $f(k)$, which is the sum of the terms in (a). (c) Response function $f(k)$ for different values of ϵ with $\omega = 0.14$ (solid lines) and with $\omega = 0$ (dashed lines). The black dashed line and crosses indicates k^* in the presence of assortativity. The other parameters are the same as in Fig. 1 of the main paper: Reduced transmissibility is $\hat{\lambda} = 2$; degree distribution is a negative binomial with mean 2.0 , coefficient of variation 4.7 .

(a) Phase diagram of PrEP in MSM communities. The x-axis shows the effective prevalence, i.e., the fraction of individuals who are living with HIV and can potentially transmit it. The y-axis shows efficacy of PrEP. The horizontal dashed line is 60% efficacy. The black curve is critical efficacy ϵ_c , the white curve is ϵ_r . While the solid lines report the results in the presence of assortativity ($\omega = 0.14$), the dashed lines report the results of the main paper (no assortativity, $\omega =$

0.0\$). The range of effective prevalence in the high-efficacy region at 60% efficacy is colored in green. The range of effective prevalence in the low-efficacy region at 60% efficacy is colored in dark blue, and light blue (transition zone). (b) & (c) Maps showing parameter region estimates in 58 countries, 24 cities with assortativity fixed as $\omega = 0.14$ (b) and $\omega = 0.28$ (c), respectively. Communities in the high-efficacy region are green, communities in the low-efficacy region are in dark blue, light blue (transition zone).

Assumptions about contact patterns in the population should be made more clear and discussed already when the model is introduced.

This is now the paragraph introducing the structure of contacts (see also previous reply):

We quantitatively investigate the existence, and phenomenology, of this trade-off, using the heterogeneous mean field formalism on an annealed network with degree distribution $p(k)$. Each node in the network has degree k sampled from $p(k)$, establishing k contacts (links) with other nodes. Heavy-tailed degree distributions are typically used to model heterogeneity in the number of contacts. We assume here that node degrees along links are not correlated. Real networks may however exhibit assortative behavior: high-degree nodes tend to be in contact with high-degree nodes. In Supplementary Note 4 we cover the case of assortative networks.

2. The authors say that using an SI framework for HIV simply means that the recovery rate can be interpreted as population turnover. However, there is a major difference between these processes. Recovery rates in the population depend on the prevalence, i.e. higher prevalence means also more recoveries per time unit. However, this should not be the case for population turnover, where more deaths from a disease would not automatically be replaced by births. The demographic process in the model is not really explained and only appears implicitly.

We have now added a section in the SI (Supplementary Note 8) to clarify this point. You are absolutely right that, in reality, recovery and turnover are two different processes. However, in the SI we now prove rigorously that if one assumes that both the size of the community, and PrEP coverage, are constant or change slowly in time, then, then the full compartmental model, containing population turnover explicitly, reduces to our SIS model, where “recovery” rate plays the role of turnover. These assumptions are reasonable as we are not interested in analyzing fluctuations happening at short time scales, as our analysis is performed “at equilibrium”.

3. In the introduction the authors say that they are using a complex networks approach. However, in networks there would be dependency between the contacts of a node, i.e. in their infection status. This is not taken into account here.

You are right in pointing this out. We did write that we use the “heterogeneous mean field formalism on an annealed network”, which is the customary method to derive the diffusion equations, without dynamical correlations. But we did not explicitly state what this formalism

entails, in terms of the form of the equations. We have now added the following paragraph, right before introducing the equations:

The heterogeneous mean-field formalism is a customary approach to write the equations describing the evolution in time of the spread of the disease in terms of the probability, by degree class, that a node is infected. It can deal with arbitrary degree distributions, while factoring out all dynamical correlations in the status of connected nodes, which would render the theory intractable.

Minor comments:

Lines 78-93: Please add some explanation of the function ξ

We have added the following sentence:

ξ is an auxiliary variable that encodes the probability of establishing a contact with an infected individual. It is the extension, to the case of partially immunized population, of the customary coupling term of the heterogeneous mean-field equations. The form of ξ given in Eq. 1 implies no degree-degree correlations: see Supplementary Note 4 for nonzero assortativity.

After equation (4): explain what z is. Is $\Phi < 1$? Are Φ and $\Psi > 0$?

Equation (6): How can ϵ_c be ≤ 1 ? From equation (6) I conclude $\epsilon_c \geq 1$.

We have added an interpretation of z in the Methods:

z has a clear epidemiological interpretation, as it measures the expected number of at-risk contacts that an individual makes. Specifically, $z = \langle k \rangle I$, where I is the probability that a given contact is with an infected individual. This measure is sensitive to the amount of heterogeneity in the network. Indeed, if the network had a homogeneous degree distribution (i.e., all individuals had degree close to $\langle k \rangle$), then $z \approx \langle k \rangle I$ (I is the prevalence as usual). Broad degree distributions give instead $z > \langle k \rangle I$, meaning that the probability of establishing a contact with an infected individual is higher than the probability of finding an infected individual at random in the population.

ϕ and ψ instead are more complex moments of the degree distribution and are harder to interpret. However, we realized that the way they appeared in the critical efficacy was misleading, since both numerator and denominator were negative. We have now rewritten the expression of ϵ_c : both terms are now positive, which makes its analysis clearer. It is now straightforward to see that $\epsilon_c \leq 1$. Additionally, we included a paragraph in the Methods with a detailed discussion under which conditions $k^* > 0$ and $0 < \epsilon_c < 1$. We show that whenever $0 < \epsilon_c < 1$, $\phi < 1$ and $z(1-\phi) > 2\psi$. Furthermore, ϕ and ψ are bigger than zero since both numerator and denominator in their definition are positive.

I find the terminology “high efficacy phase” a bit confusing, because I with the word “phase” I connect something that changes over time. I suggest the term “high prevention efficacy”.

Yes, we see that "phase" might be misleading, as one would think of the phase of an epidemic. We have thus replaced "phase" with "parameter region" or "region", when relevant.

Lines 116-118: this needs some more explanation. How does the underlying contact structure play a role here? Maybe it would be good to mention the contact structure already when the model is introduced, and say how it influences the prevalence.

We realized that this sentence is actually redundant, as in the paragraphs that follow we examine the impact of both prevalence and contact structure. In this sense we see how the sentence is not clear, as it hints at something we discuss later, without providing additional details. We thus decided to drop it altogether. Also, we now describe the contact structure when we introduce the model (see previous replies).

Line 192: "these" instead of "this"

Done. Thank you for spotting the typo.

Line 197: Unclear how treatment is included in the model. Treatment would lower the infectiousness of those under treatment or would move them back to the susceptible state. This is now not included in the model.

We have added a section to the Supplementary Information (Supplementary Note 8) in which we explain in detail the impact of treatment in our model. We write down the equations for a model which explicitly describes both treatment and population turnover. Then, we prove that, if treatment coverage is constant in time, or at least varying slowly, then the role of treatment is to effectively lower the transmissibility λ , which becomes $\lambda(1-r)$, with r being the recovery rate. The assumption of constant treatment coverage is reasonable because we are not interested in fluctuations happening at short time scales, as we study the system "at equilibrium". Our proof shows that treatment also changes the effective recovery rate, which is required to keep the population constant.

Line 235: Here the number of individuals using prevention is mentioned, but up to there the model for formulated in terms of fractions, and also the prevalence I as defined in line 85 is a fraction. This seems inconsistent to me. Please explain exactly how I is defined as a function of numbers of people taking prevention.

We realize now that the explanation in that part of the Methods may be misleading. " I " is consistently defined as prevalence (fraction). But if you derive prevalence directly by g_k (fraction of those on PrEP), you would actually be comparing offering PrEP to a different number of people, across different degree classes. This is because a change of 1% in g_k in a degree class with 1,000 people means 10 additional people on PrEP. Instead, the same change (1%) in a degree class with 100 people means only one additional person on PrEP. This is why we derive by $(N p_k g_k)$, which is the number of people with degree k and on PrEP, in a population of N individuals. To avoid the original ambiguity, we now also multiply the numerator by N (NI), to have absolute numbers both at the numerator, and at the denominator. Then, of course, N simplifies, as the final result does not depend on population size, and one gets the final definition of the response function.

The relevant part in the Methods now reads

The goal of $f(k)$ is to measure the impact that providing prevention to few individuals in degree class k has on community-level baseline prevalence. We assume a population of N individuals, and define f as the change in the number of infected individuals in the population (NI), due to a small change in the amount of prevention provided in degree class k ($Np_k g_k$):

$$f(k) = - \frac{d(NI)}{d(Np_k g_k)} \Big|_{g=0}$$

where the minus sign is due to the fact that prevention will bring prevalence down. Here I is community-level prevalence as defined previously. Then, the population size N correctly cancels out (the final result does not depend on population size), and we get to the final definition of the response function:

$$f(k) = - \frac{1}{p_k} \frac{dI}{dg_k} \Big|_{g=0}$$

Reviewer #2

The manuscript deals with the interesting idea that protecting the individuals with the highest contact rates is not always the best strategy to reduce the prevalence of an endemic disease when prevention is not 100% effective (leaky prevention). The situation the authors have in mind is the use of pre-exposure prophylaxis (PrEP) in control HIV and they apply their results using HIV epidemiological data from MSM communities in 58 countries and 24 cities.

To prove this claim, the epidemic spread is modelled using a heterogeneous mean-field SIS model with two types of infected individuals: those who do not receive prevention and those who do. The goal is then to find the optimal distribution of PrEP among individuals with different degrees that maximizes the initial reduction of the prevalence when PrEP are distributed in a population where no preventive measures are adopted.

As far as I know, the manuscript offers a new approach to deal with endemic infectious diseases and presents new insights of significance to the field of epidemics. The paper is, in general, well written but some revision is needed.

First, the explanation of the model could be better if it were done with a general audience in mind, i.e., readers unfamiliar with network theory. In particular, an interpretation of ξ (or, better, $\xi / \langle k \rangle$) would help to better understand of the model because it is the key ingredient.

We now include a more detailed explanation of the underlying contact pattern when introduce the model:

We quantitatively investigate the existence, and phenomenology, of this trade-off, using the heterogeneous mean field formalism on an annealed network with degree distribution $p(k)$. Each node in the network has degree k sampled from $p(k)$, establishing k contacts (links) with other nodes. Heavy-tailed degree distributions are typically used to model heterogeneity in the number of contacts. We assume here that node degrees along links are not correlated. Real networks may however exhibit assortative behavior: high-degree nodes tend to be in contact with high-degree nodes. In Supplementary Note 4 we cover the case of assortative networks.

Regarding ξ , we have added the following sentence:

ξ is an auxiliary variable that encodes the probability of establishing a contact with an infected individual. It is the extension, to the case of partially immunized population, of the customary coupling term of the heterogeneous mean-field equations. The form of ξ given in Eq. 1 implies no degree-degree correlations: see Supplementary Note 4 for nonzero assortativity.

Moreover, it is important to mention the assumptions of this formulation (for instance, it is assumed that nodal degrees are uncorrelated).

Thank you for raising this point. Prompted by your comment, and by the other reviewers, we now include a full treatment of degree-degree correlations (assortativity).

In the SI we now include a section (Supplementary Note 4) in which we derive the efficacy thresholds ϵ_c , ϵ_r , and the optimal degree k^* , for arbitrary assortativity. The approach mixes analytical derivation, and numerical evaluation of formulas which cannot be simplified analytically.

We find that assortativity increases the critical efficacy ϵ_c , increases ϵ_r , and, in the low-efficacy region, decreases the optimal degree k^* . This happens because assortativity increases risk of exposure among those already at high risk (many contacts). This in turn increases the probability of getting infected despite prevention.

We then apply these new findings to the case of PrEP, and re-assign the communities in Fig. 3 to the different efficacy regions, assuming a value of assortativity that was reported for Sweden (Fig. S4a,b). Assortativity causes no change in region assignment for most communities: only 4 go from the high-efficacy region (assuming no assortativity), to the transition zone (with assortativity). Plus, this entails no change in strategy, given that both the high-efficacy region and the transition zone risk-based distribution is recommended. We also tested a value of assortativity twice as high as that, to explore communities with mixing patterns which might be very different from the Swedish study, which can be hardly regarded as representative of different socioeconomic and epidemiological contexts. In this case, compared to the analysis in the main text (no assortativity), 7 communities moved from the high-efficacy region to the transition zone, and 2 communities moved from the transition zone to the low-efficacy zone. This shows that assortativity may change recommendations for PrEP distribution only at extremely high values (i.e., those at high risk strongly favoring mixing with others at high risk), and for few communities. Notwithstanding, further studies applied to specific settings would certainly benefit from community-specific estimates of partner selection patterns (and the resulting assortativity), in order to accurately estimate ϵ_c and ϵ_r , in case assortativity turns out to be very different from the values tested.

We have added the following paragraphs to the main text:

1.

We quantitatively investigate the existence, and phenomenology, of this trade-off, using the heterogeneous mean field formalism on an annealed network with degree distribution $p(k)$. Each node in the network has degree k sampled from $p(k)$, establishing k contacts (links) with other nodes. Heavy-tailed degree distributions are typically used to model heterogeneity in the number of contacts. We assume here that node degrees along links are not correlated. Real networks may however exhibit assortative behavior: high-degree nodes tend to be in contact with high-degree nodes. In Supplementary Note 4 we cover the case of assortative networks.

2.

Degree-degree correlations also increase the critical efficacy ϵ_c , and, in the low-efficacy region, decrease k^ (see Supplementary Note 4). Intuitively, this happens because assortativity increases risk of exposure among those already at high risk, exacerbating the likelihood of breakthrough infections.*

3.

We also tested the impact of assortative mixing, which is reported in many MSM communities. Namely, location-based partner selection, and homophily, may cause those at high risk mixing preferably with other high-risk individuals. Assortativity had little effect on the efficacy estimates of Fig. 3b: Specifically, with the assortativity estimated in Ref. 4, only 4 out of 34 communities moved from the high-efficacy region to the transition zone, with risk-based distribution still outperforming non-selective distribution (see Supplementary Note 4). We also checked a value of assortativity twice as much as that of Ref. 4 (see Supplementary Note 4): in that case, 7 of 34 communities moved from the high-efficacy region to the transition zone, and 2 out of 4 moved from the transition zone to the low-efficacy region. This shows that assortativity may change recommendations for PrEP distribution only at extremely high values (i.e., those at high risk strongly favoring mixing with others at high risk), and for only 2 out of the 76 communities investigated here.

The new section in the SI (Supplementary Note 4 - see in particular Figures S3 and S4) describes the computation of $f(k)$, ϵ_c , ϵ_r , k^* . It also further illustrates the effect of assortativity on the optimal choice of PrEP distribution strategies.

Second, the main theoretical result -Eq.(6)-, is given in the main text without the meaning/interpretation of any of the ingredients, even though at least one of them (z) is easily interpretable. From a modelling point of view, the meaning and/or properties of the averages ψ and ϕ , the other two ingredients, seem to be very important. For instance, if $\phi \geq 1$, the optimal degree k^* given by Eq. (5) is negative. In such a case, what does $k^* < 0$ mean? Is this hypothesis on ϕ feasible?

On the other hand, if $\phi < 1$, then $k^* > 0$ if the denominator is positive but, in this case, the critical efficacy given by the right-hand side of Eq. (6) will be negative if $2\psi > z(1-\phi)$ (the positivity of the denominator of k^* already guarantees that $z(1-\phi) > \psi$). So, I think that a more detailed discussion about the conditions that guarantee $k^* > 0$ and critical efficacy < 1 would be very helpful.

You are right that z is easily interpretable, and we now provide its interpretation in the Methods:

z has a clear epidemiological interpretation, as it measures the expected number of at-risk contacts that an individual makes. Specifically, $z = \langle k \rangle I$, where I is the probability that a given contact is with an infected individual. This measure is sensitive to the amount of heterogeneity in the network. Indeed, if the network had a homogeneous degree distribution (i.e., all individuals had degree close to $\langle k \rangle$), then $z \approx \langle k \rangle I$ (I is the prevalence as usual). Broad degree distributions give instead $z > \langle k \rangle I$, meaning that the probability of establishing a contact with an infected individual is higher than the probability of finding an infected individual at random in the population.

ϕ and ψ instead are more complex moments of the degree distribution and are harder to interpret. Notwithstanding, thanks to your comment, we realized that the way they appeared in the critical efficacy was misleading, since both numerator and denominator were negative. We have now rewritten the expression of ϵ_c : both terms are now positive, which

makes its analysis clearer. Additionally, we included a paragraph in the Methods with a detailed discussion under which conditions $k^* > 0$ and $0 < \epsilon_c < 1$. We show that whenever $0 < \epsilon_c < 1$, $\phi < 1$ and $z(1-\phi) > 2\psi$. Furthermore, ϕ and ψ are bigger than zero since both numerator and denominator in their definition are positive.

Moreover, there are very important claims like “highly prevalent diseases have higher ϵ_c , and, in the low-efficacy phase, lower k^* ” and “more heterogeneous contact networks have higher ϵ_c ” that are not justified. Are these statements obtained only from the numerical simulation of the model? Can they be proved from the definition of ϵ_c and the properties of ψ , ϕ , and z ? What is the relationship between these quantities and disease prevalence?

Our original analysis was exclusively numerical. Prompted by your comment, we have derived also the analytical proof of these statements, which is now a section in the Methods. Thank you for the suggestion!

Finally, it would be interesting to confirm numerically that the prevalence in the new endemic equilibrium that emerges after the introduction of preventive measures actually follows the predictions based on the linear response function.

We have added a section in the SI (Supplementary Note 3), where we numerically confirm the theoretical prediction of the linear response function. To be more precise, we iteratively introduce a small fraction of individuals on PrEP in degree class k and solve the differential equations of the epidemic models. Accordingly, the function $f(k)$ is then calculated through finite differences considering prevalence before and after the introduction of PrEP. Similarly, we evaluated $F_{\text{indir}}(k)$ by calculating numerically the derivative dx_m/dg_k and averaging the terms according to the degree distribution. Finally, $F_{\text{dir}}(k)$ is directly calculated through $y_k - x_k$. The results are shown in Fig. S2 of the SI. The agreement between theory and the numerical results is almost perfect.

Other comments:

Why a negative binomial degree distribution is used as a main example? It is well known that distributions of contacts with high variance (negative exponential, truncated scale-free distribution) are more suitable to model contact patterns among individuals?

In the manuscript we analyze both a negative binomial distribution (main), and a power law (Supplementary Note 2). We agree with the reviewer that many distributions to model sexual contacts are possible, and many have been used in the literature. See, for instance, Hamilton et al. 2008 (https://journals.lww.com/stdjournal/Fulltext/2008/01000/Degree_Distributions_in_Sexual_Networks_A.8.aspx), where they compared different models, including negative binomial and power law. Hancock et al. 2004 (<https://www.sciencedirect.com/science/article/pii/S0040580904000310?via%3Dihub>) tested five datasets and found the negative binomial model to be the best in four of them, the power law in the remaining one.

In addition their widespread use, the reason why we chose these two particular distributions is that they are sometimes viewed as "competing" models of human contacts, with the power laws coming from the field of social sciences, and the negative binomial from the field of applied statistics, as a means to add overdispersion to a Poisson process. By testing both families of distributions, our goal is to show that our findings on the impact of contact heterogeneity are robust whatever the choice of the underlying distribution.

Moreover, the example of COVID-19 vaccination considers such a degree distribution with a mean degree equal to 1, which seems to be quite unrealistic.

We thank the referee for pointing this out. We adapted the example of COVID-19 and fixed the mean degree as 13.4, which is the average number of daily contacts inferred in Ref. 30 of the SI. The conclusion remains that COVID-19 does not fall into the low-efficacy region.

The definition of $F_{dir}(k)$ given in (2), namely, $(y_k - x_k)|_{g=0}$, leads to Eq. (3) multiplied by -1 when it is used in combination with Eqs. (9) and (10). I think its definition should be $(x_k - y_k)|_{g=0}$.

You are right, thank you for spotting that. Corrected.

The writing of the section "Invasion stage and epidemic threshold" has to be carefully revised (see last minor comments below).

We have restructured this section, to provide a clearer explanation of the derivation of the epidemic threshold.

In summary, although the manuscript is interesting and offers novel insights, I cannot recommend its publication in its current form but I encourage the authors to improve the writing.

Thank you for your input. We have added the required analysis, and restructured the writing following your suggestions.

Minor comments:

- The use of the name "diffusion equations" to refer to the system of ODEs governing the dynamics of the disease is misleading and it is not the usual one.

We have rephrased. It now reads

The heterogeneous mean-field formalism is a customary approach to write the equations describing the evolution in time of the spread of the disease in terms of the probability, by degree class, that a node is infected.

- Eq.(10) directly follows from the equilibrium equation for (1) after replacing ξ by z since $\xi(g=0) = z$. I don't see why Ref.[42] is needed.

Well spotted, we changed it.

- Line 103: $F_{\text{indir}}(k)$ does not decrease but increases with k .

Corrected.

- Line 108: "... it is the result of..."

Corrected.

- Line 133: Is the sentence " ϵ_c decreases as overdispersion increases" correct? In Fig.2b, ϵ_c increases with the coefficient of variation.

It was a typo. Now corrected to " ϵ_c increases as overdispersion increases"

- Line 249: I think the use of "by" in the expression "we derive the first line in Eq (1) by g_m " is not correct in English.

We realized that our phrasing was ambiguous. We meant "to derive by" in the sense of performing a derivative. We have rephrased as follows:

we perform the derivative $\frac{d}{dg_m}$ on both sides of the first line in Eq. (1)

- Check the left-hand side of Eq.(14). The second term in parentheses must be $(1 + \hat{\lambda} z k)$

Thank you, you are right. Corrected.

- Line 267: Give more detail in the sentence "The condition $\lim_{k \rightarrow \infty} f'(k) < 0$ leads to ϵ_c of Eq.(6)."

We thank you for pointing this out. We adapted this section and it now includes a detailed derivation of how the condition $\lim_{k \rightarrow \infty} f'(k) < 0$ leads to ϵ_c .

- The linearization about the disease-free state of the first equation in (1) is equal to the right-hand side of Eq.(20) multiplied by μ .

Correct. We have restructured the section, and highlighted how the derivation of the epidemic threshold comes from linearizing Eq. 1 around disease-free state.

- Check the expression (21) of Θ , in particular, the factor $\langle k \rangle$ in the sum.

Now restructured. See previous comments.

- Line 274: Specify the variable with respect to which the derivative is to be taken.

Now restructured. See previous comments.

- The denominator in the fraction appearing in (23) has to be $\langle k^2 \rangle$ instead of $\langle k \rangle$.

Corrected, thanks.

Reviewer #3

This manuscript explores the impact of current targeted PrEP recommendations, with a discussion of when non-selective PrEP distribution may be optimal. I have a few questions/recommendations for the authors:

- The authors discuss alternative dosing strategies and adherence, but now long-acting injectable PrEP is available in certain countries. How could this impact calculations and estimates?

Thank you for suggesting this point of discussion: this is indeed very relevant. We have added the following paragraph to the text:

Finally, the availability of new PrEP formulations may affect the conditions and timing of transition to the high-efficacy region. Notably, long-acting injectable cabotegravir (CAB-LA) was recently shown to have higher efficacy than oral PrEP (Landovitz et al 2021 NEJM). This means that communities that are now in the low-efficacy region for oral PrEP, may be in the high-efficacy region for CAB-LA.

- I wonder whether the authors have considered some important characteristics in HIV risk beyond PrEP adherence and number of sexual interactions. If not, these should be included in the Limitations section. Specifically, it would be useful to at least mention:

--- Differences in per-act risk (receptive vs. insertive anal sex)

Indeed our model does not include the difference in transmissibility between receptive and insertive anal sex. We have edited the limitations as follows:

The compartmental model we used is a coarse-grained representation of the progression of HIV infection, and its transmission. In particular, it does not account for the different transmission probability of receptive and insertive anal sex. This, however, would potentially bias our findings only if PrEP use were consistently correlated with type of act (insertive vs receptive). Summing up, more detailed HIV models, and community-specific estimates of partner selection patterns, could provide better numerical estimates of critical efficacy, and thus be useful in applied studies focusing on specific communities.

--- Prevalence of condom usage (which could help counter poor adherence)

We have rephrased the relevant paragraph of the limitations as follows, to discuss this:

Our study did not include factors which can influence risk of acquisition: in the case of HIV, we did not explicitly account for the effect of primary prevention other than PrEP. Specifically, whereas our framework does account for an arbitrary overall rate of condom use by means of the transmissibility parameter λ , it does not include possible changes in condom use among those on PrEP, due to possible behavioral adaptation.

--- Impact of criminalization of same-sex behavior. For example, sexual negotiation for MSM in Nigeria is very different than in Canada.

Good point. Stigma and criminalization are likely associated with a decrease in consistent PrEP use, leading to lower PrEP efficacy. This means that they are contributing factors to being in the low-efficacy parameter region. We have added the following paragraph to the discussion:

Stigma and criminalization of same-sex acts are other factors possibly associated with the low-efficacy region: They are obstacles to PrEP use, as they make it harder to supply the medication, and to provide consistent support and follow up. This decreases adherence, which in turns decreases efficacy. Decriminalization and societal changes leading to lower stigma may thus signal a transition from the low-efficacy to the high-efficacy region.

--- Role of homophily in transmission.

Thank you for raising this point. Indeed your comment, and those of the other reviewers, prompted us to add to our analysis homophily, and other partner selection mechanisms, which can induce degree-degree correlations (assortativity) in the contact network.

We have added this paragraph to the manuscript, to discuss the role of homophily:

We also tested the impact of assortative mixing, which is reported in many MSM communities. Namely, location-based partner selection, and homophily, may cause those at high risk mixing preferably with other high-risk individuals. Assortativity had little effect on the efficacy estimates of Fig. 3b: Specifically, with the assortativity estimated in Ref. 4, only 4 out of 34 communities moved from the high-efficacy region to the transition zone, with risk-based distribution still outperforming non-selective distribution (see Supplementary Note 4). We also checked a value of assortativity twice as much as that of Ref. 4 (see Supplementary Note 4): in that case, 7 of 34 communities moved from the high-efficacy region to the transition zone, and 2 out of 4 moved from the transition zone to the low-efficacy region. This shows that assortativity may change recommendations for PrEP distribution only at extremely high values (i.e., those at high risk strongly favoring mixing with others at high risk), and for only 2 out of the 76 communities investigated here.

We have added a section to the Supplementary Information (Supplementary Note 4) in which we derive the effect of assortativity, and we describe how we estimate its impact on our findings.

- Stating that non-selective PrEP distribution is optimal is helpful, but I would like the authors to go a bit further with their recommendations. What does this look like? Several thresholds are mentioned for high vs. low efficacy - what is recommended for the transition from non-selective to targeted?

We have rephrased this discussion paragraph, and added the relevant comment:

Non-selective PrEP distribution is effective when HIV prevalence is high and/or treatment coverage low. Then, as prevalence goes down and treatment increases, focusing on protecting individuals at highest risk will likely become the best-performing strategy. At the same time, more consistent use of oral PrEP, or new long-acting PrEP formulations may speed up the progression to the high-efficacy region. When this happens, it is possible that many communities will find themselves in the transition zone, at least temporarily. There, risk-based distribution should already be favored over non-selective distribution, as in the high-efficacy region.

REVIEWERS' COMMENTS

Reviewer #1 (Remarks to the Author):

Thank you for your extensive answers to my comments and the additional analyses you provided. I am happy with these answers and recommend the paper for publication.

Reviewer #2 (Remarks to the Author):

The revised version of the manuscript addresses all points in my report. Moreover, some parts of it have been rewritten improving its comprehension. However, I still have a small question concerning the modeling assumptions.

The authors have partially responded to the first point of my report, which suggested that the explanation of the model should be addressed to a more general audience. In this regard, I think it is important (and still missing) to explain the meaning of considering an annealed network and the assumption underlying that choice, namely that changes in the network are assumed to occur much faster than the spread of the disease (time scale separation). Why are these networks more appropriate to model the spread of HIV than static/quenched networks? Is this fact related to an alleged promiscuity in the communities under consideration?

In conclusion, I recommend the publication of the manuscript in Nature Communications provided that the authors address the previous questions.

Minor comment:

Use "Eqs.(x)-(y)" instead of "Eq.(x),(y)" when referring to more than one equation.

Reviewer #3 (Remarks to the Author):

The authors have adequately addressed my specific concerns. I have no further comments/edits.

MANUSCRIPT
NCOMMS-21-48514A

TITLE

Non-selective distribution of infectious disease prevention may outperform risk-based targeting

Point-by-point response to reviewers' comments

Reviewer #1

Thank you for your extensive answers to my comments and the additional analyses you provided. I am happy with these answers and recommend the paper for publication.

Thank you for the careful review of our work and the positive evaluation received. The comments led to a much improved manuscript.

Reviewer #2

The revised version of the manuscript addresses all points in my report. Moreover, some parts of it have been rewritten improving its comprehension. However, I still have a small question concerning the modeling assumptions.

The authors have partially responded to the first point of my report, which suggested that the explanation of the model should be addressed to a more general audience. In this regard, I think it is important (and still missing) to explain the meaning of considering an annealed network and the assumption underlying that choice, namely that changes in the network are assumed to occur much faster than the spread of the disease (time scale separation). Why are these networks more appropriate to model the spread of HIV than static/quenched networks? Is this fact related to an alleged promiscuity in the communities under consideration?

Thank you for giving us the chance to clarify this. We believe the annealed network formalism is the optimal choice in this context, for two reasons:

1. You correctly mention 'time scale separation'. The time scale of HIV transmission in MSM communities is, on average, orders of magnitude larger than the time scale of network evolution. We can get a rough estimate of these time scales as follows. We focus on the type of links in the network that change at the fastest pace: one-time partnerships, which, according to Ref. [8] (of the manuscript), have a mean rate of acquisition of $r_{network} = 0.16 \text{ week}^{-1}$. Thus, the lower bound of the time scale of network evolution is $\tau_{network} \approx 6 \text{ week}$. Then, we focus on the contact at highest risk of HIV acquisition: unprotected, receptive anal intercourse, which has a per-act probability of transmission of $p = 0.014$ [Baggaley et al.]. Thus, the lower bound of the time scale of disease transmission is $\tau_{disease} = (p r_{network})^{-1} \approx 450 \text{ week} \gg \tau_{network}$. This proves the time scale separation. Also, the estimate of $\tau_{disease}$ is indeed a lower bound, because it assumes that all acts are highest-risk acts, and that all partners may spread HIV: the actual $\tau_{disease}$ is thus probably much higher, making $\tau_{disease}$ even higher.
2. Reconstructing the exact (quenched) contact network in MSM communities is beyond the power of surveys. That would require cohorts with extremely high adherence, and response rate (given the above argument about time scale separation, the exact network structure would need to be sampled very frequently). This is unattainable, especially in light of the amount of stigma and marginalization of these communities in many countries, making them hard-to-reach. Also, knowing the exact network structure would not improve mathematical models, given that we are in the regime where the annealed network formalism works well anyways (see point 1). This is why most mathematical models of HIV among MSM assume - explicitly, or implicitly - an annealed network. It is the case, for instance, of Ref. [5] that we cite in the manuscript, where the authors divide the population into discrete groups featuring different sexual activity.

Ref:

Baggaley R. F., et al (2010), HIV transmission risk through anal intercourse: systematic review, meta-analysis and implications for HIV prevention, International Journal of Epidemiology 39, 4 1048–1063.

We have added the following paragraph to the manuscript:

Annealed networks are particularly suitable when the timescale of pathogen spread is much larger than the timescale at which contacts change¹⁶, as is the case of HIV epidemics in MSM communities (Supplementary Note 2). Also, annealed networks can be parametrized from existing surveys^{5,13,32}, unlike more complex network models, which would require high-resolution contact data.

In Supplementary Note 2 we now outline the proof, as explained above, of $\tau_{disease} \gg \tau_{network}$ for HIV epidemics in MSM communities.

In conclusion, I recommend the publication of the manuscript in Nature Communications provided that the authors address the previous questions.

Thank you for the careful review of our manuscript. Your suggestions have been very helpful in improving our study.

Minor comment:

Use "Eqs.(x)-(y)" instead of "Eq.(x),(y)" when referring to more than one equation.

Thank you for spotting that, the manuscript now follows your recommendation.

Reviewer #3

The authors have adequately addressed my specific concerns. I have no further comments/edits.

Thank you for the careful review of our manuscript and the precious suggestions provided.